# Hospital Wastewater Surveillance and Antimicrobial Resistance: A Narrative Review

**DOI:** 10.3390/microorganisms13122739

**Published:** 2025-11-30

**Authors:** Diamantina Lymperatou, Revekka Konstantopoulou, Michalis Mentsis, Natalia Atzemoglou, Christina Diamanti, Ioannis Tzourtzos, Katerina K. Naka, Michail Mitsis, Gartzonika Konstantina, Haralampos Milionis, Evangelia Ntzani, Eirini Christaki

**Affiliations:** 1First Division of Internal Medicine & Infectious Diseases Unit, University General Hospital of Ioannina, Faculty of Medicine, University of Ioannina, 45500 Ioannina, Greece; andalimperatou@gmail.com (D.L.); revekkankon@gmail.com (R.K.); md06737@uoi.gr (M.M.); md06684@uoi.gr (N.A.); hmilioni@uoi.gr (H.M.); 2Department of Hygiene and Epidemiology, Faculty of Medicine, University of Ioannina, 45500 Ioannina, Greece; ch.diamanti@uoi.gr (C.D.); entzani@uoi.gr (E.N.); 3Second Department of Cardiology, University General Hospital of Ioannina, Faculty of Medicine, School of Health Sciences, University of Ioannina, 45500 Ioannina, Greece; ioannistzourtz@gmail.com (I.T.); anaka@uoi.gr (K.K.N.); 4Department of General Surgery, University General Hospital of Ioannina, Faculty of Medicine, University of Ioannina, 45500 Ioannina, Greece; mmitsis@uoi.gr; 5Department of Microbiology, University General Hospital of Ioannina, 45110 Ioannina, Greece; kgartzon@uoi.gr; 6Centre for Evidence Synthesis in Health, School of Public Health, Brown University, Providence, RI 02912, USA; 7Biomedical Research Institute, Foundation for Research and Technology, 45110 Ioannina, Greece

**Keywords:** antimicrobial resistance, hospital wastewater surveillance, antimicrobial resistance genes, antimicrobial stewardship

## Abstract

Antimicrobial resistance (AMR) poses a critical global health threat. Wastewater surveillance has recently emerged as a valuable public health tool for monitoring AMR in communities and healthcare settings. This narrative review comprehensively examines the role of hospital wastewater surveillance (HWWS) in monitoring antimicrobial resistance. Methods to detect resistant bacteria and antimicrobial resistance genes (ARGs) in wastewater systems, ranging from culture-based techniques to advanced molecular approaches, including polymerase chain reaction (PCR) and next-generation sequencing (NGS), are explored. The review synthesizes data on key antimicrobial resistance genes commonly detected in hospital effluents and explores how HWWS contributes to understanding the dynamics of resistance within healthcare settings. Furthermore, it identifies methodological challenges and inconsistencies in data reporting and outlines necessary standardization steps to enhance the effectiveness of HWWS programs. Opportunities for integrating HWWS with clinical and public health frameworks are presented, emphasizing the need for robust metadata and transparent reporting. This review provides a comprehensive approach to HWWS strategies, which could complement robust infection control and antibiotic stewardship interventions to combat AMR.

## 1. Introduction

Antimicrobial resistance (AMR) has emerged as one of the most pressing global public health challenges, with projections indicating that it will worsen significantly in the upcoming decades if urgent and targeted interventions are not implemented [1]. In 2021, AMR caused an estimated 1.14 million deaths globally and was associated with approximately 4.71 million additional deaths [2]. AMR transmission has been observed to occur through direct contact between hospital settings and patients. In particular, hospital wastewater (HWW) has been identified as a potential reservoir for pathogens that can lead to human infections acquired within the environment [3]. Beyond human health, AMR threatens water safety, agriculture and ecology, making it a paradigm for a One Health issue.

The challenge of AMR has become more prominent during recent decades, also because of the slow rate of development of novel antimicrobials. Notably, most new agents represent modifications to existing drug classes rather than antimicrobial classes with new mechanisms of action. While AMR knows no borders and affects every country, low- and middle-income countries may bear the greatest burden, as they suffer from limited access to advanced-generation antimicrobials, weaker infection control infrastructure, and environmental mismanagement of pharmaceutical waste—factors that illustrate the complexity of AMR [1].

In 2020, Europe alone reported more than 800,000 infections caused by drug-resistant bacteria, resulting in more than 35,000 attributable deaths [4]. The 2022 European Centre for Disease Prevention and Control (ECDC) annual report described persistently high resistance rates among clinically important bacterial pathogens: 14.3% of *Escherichia coli* (*E. coli*) isolates displayed resistance to third-generation cephalosporins, and 22% to fluoroquinolones. Among *Klebsiella pneumoniae* isolates, 10.9% exhibited carbapenem resistance. Additionally, for *Pseudomonas aeruginosa* (*P. aeruginosa*), 19.3% of isolates were resistant to piperacillin/tazobactam, while 15.2% of *Staphylococcus aureus* (*S. aureus*) isolates demonstrated methicillin resistance. The resistance rates in *Acinetobacter* spp. reached 36.3% for carbapenems across Europe. Regarding Greece, AMR rates remain significantly higher than the EU population-weighted average, meaning resistance to fluroquinolones in *E.coli* reaches 37.8%, *P. aeruginosa* resistance to piperacillin/tazobactam stands at 50.5%, *methicillin-resistance S. aureus* (*MRSA*) at 39%, and *Acinetobacter* spp. carbapenem resistance at 95.9% [5]. This profound AMR burden in Greece underscores the urgency of developing innovative surveillance frameworks aiming to monitor emergence pathways and environmental dissemination routes.

In this context, wastewater surveillance (WWS) has emerged as a promising complementary approach to tracking AMR. Although WWS gained significant prominence during the SARS-CoV-2 pandemic, it has been used for decades to track infectious diseases at the population level [6]. In the context of AMR surveillance, this environment-based approach addresses many limitations of traditional event-based methods: it does not rely on cases of illness involving drug-resistant pathogens, does not require healthcare access, avoids privacy issues, and can reduce costs [7,8,9]. The abundance of ARGs is believed to be an important indicator of AMR pollution in different environments, and the ARGs enriched in hospital wastewater (HWW) systems may reflect the AMR distribution in clinically related environments [10].

A specific and particularly relevant subset of WWS is hospital wastewater surveillance (HWWS), which focuses on the effluent generated within healthcare facilities. According to the World Health Organization (WHO), hospital wastewater refers to water discharged from all hospital-related activities, both medical and non-medical, including effluent from surgery rooms, diagnostic laboratories, patient wards, laundries, and kitchens. HWW may contain a mixture of pathogenic microorganisms, antibiotic residues, resistant bacteria strains, heavy metals, and chemical agents from disinfectants and pharmaceuticals. This complex composition makes HWW an important environmental reservoir for ARGs and a potential vector of AMR dissemination if inadequately treated prior to release into municipal systems [11].

Advances in detection technologies have led to great progress in the monitoring capacity of HWWS programs. Methods range from traditional culture-based techniques to advanced molecular tools such as polymerase chain reaction (PCR), next-generation sequencing (NGS), and microarray-based platforms. Each technique has its own advantages and disadvantages. For example, PCR offers rapid detection of known ARGs, and NGS provides a broad genomic preview, including novel resistance elements. The integration of those techniques into public health systems enables them to assess intervention efficacy and provide early warnings for potential outbreaks [12,13].

This narrative review aims to synthesize and discuss the existing literature on HWWS in the context of AMR, focusing on the most frequently identified ARGs and the methods to detect them, and explores the potential of HWWS to improve infection control and reduce the spread of AMR, highlighting current challenges and providing insights for further research.

## 2. Materials and Methods

This narrative review aims to synthesize and interpret published evidence regarding the presence and characterization of ARGs in HWW, as well as the methodologies employed to detect and analyze them. This review is not a systematic review or meta-analysis, as no systematic or quantitative data aggregation, meta-analysis, or meta-data analysis was performed, but follows a structured literature selection to ensure comprehensive coverage of the topics addressed.

The literature search was conducted using the PubMed database, due to its coverage of high-quality, peer-reviewed biomedical literature relevant to our research question. A total of 96 articles were initially retrieved from PubMed, and after reviewing the abstracts, 42 studies were ultimately selected for inclusion. Articles from other major electronic databases, like Scopus and Google scholar, were manually researched, and the latest reports from CDC and WHO were also included.

Search terms were designed to capture articles relevant to both hospital wastewater surveillance and antimicrobial resistance; therefore, keyword combinations included “antimicrobial resistance and resistant”, “AMR”, “antimicrobials”, “antibiotics”, and “hospital wastewater surveillance”. Boolean operators (AND, OR) were used to refine the queries. The search strategy was adapted for each database according to the library’s syntax requirements.

The inclusion criteria included articles that reported original data and structured reviews on hospital wastewater surveillance and AMR and were published in English between 2018 and 2025. This time period was chosen, as the field of wastewater surveillance has expanded after the COVID-19 pandemic, both in terms of applications and methodology. Articles that lacked methodological description or used purely theoretical modeling and non-peer reviewed materials were excluded. Given the heterogeneity of study designs, sampling protocols, and reporting measures, a narrative synthesis approach was employed.

## 3. Antimicrobial Resistance Genes That Are Identified in Hospital Wastewater

Key antimicrobial resistance genes detected in clinical isolates and hospital wastewater include those associated with extended-spectrum *beta-lactamases* (*ESBLs*) such as *bla CTX-M*, *carbapenemases* including *bla_KPC*, *bla_NDM*, and *bla_OXA* variants, and colistin resistance genes (*mca-1* and variants), as well as vancomycin resistance genes such as *vanA* and *vanB*, reflecting the critical threat posed by multidrug-resistant pathogens [3,6,14,15,16,17,18,19,20]. In a study that collected public metagenomic datasets of 71 hospital wastewater samples from 18 hospitals in 13 cities, the results revealed that the primary hosts of ARGs within HWS were found to be *Escherichia coli* and *Klebsiella pneumoniae* [10]. In this review, we primarily examined both critical and common resistance traits detected in hospital isolates, such as those related to *ESBL*, *carbapenemases*, *colistin resistance*, and *vancomycin resistance*, and we also included those associated with tetracycline and sulfonamide resistance. Most common ARGs detected in HWW and their respective mechanisms of action are included in Table 1.

### 3.1. Carbapenem Resistance Related Genes

Carbepenem resistance in *Enterobacterales* is primarily driven by the production of *carbapenemases*. These enzymes are capable of hydrolyzing nearly all beta-lactam antibiotics, which greatly limits treatment options [21]. Despite geographic variation, the most commonly encountered *carbapenemases* among the *Enterobacterales* are the Ambler class A *serine carbapenemase*, *KPC*, the class B *metallo-β-lactamases* (*MBLs*), including *NDM*, *IMP*, and *VIM*, and the class D *OXA-48-like* enzymes [22]. The *kpc* gene, encoding the enzyme *Klebsiella pneumoniae carbapenemase*, is a widespread driver of carbapenem resistance among *Enterobacteriaceae.* The detection of *kpc* in hospital wastewater often indicates a high load of resistant bacteria, a situation frequently tied to the overuse and misuse of antibiotics, as well as suboptimal infection control implementation. Research shows that *kpc* genes are commonly found in hospital wastewater samples, underscoring the value of wastewater surveillance for tracing AMR patterns [23]. The *blaKpc* is primarily associated with *Klebsiella* spp., *Enterobacter* spp., and *Escherichia coli* and has been found in other *Enterobacteriaceae*, *Acinetobacter baumannii*, *Pseudomonas aeruginosa*, as well as *Aeromonadaceae* [14,15,16,20,24,25]. Quantitative real-time PCR (qPCR) and digital PCR (dPCR) analysis of hospital wastewater has also targeted several other *carbapenemase* genes like *blaNDM*, *blaIMP*, *blaVIM*, *blaOXA-23-like*, *blaOXA-48-like*, and *blaOXA-58-like* [26]. The *blaImp* gene has been observed in Gram-negative bacteria and *Enterococcaceae* [19], while *blaVIM-1* has been detected in *Pseudomonas species and E. coli* [17]. The *blaNDM-1* gene, which also confers broad beta-lactam resistance, has been identified in *Enterobacteriaceae*, *Pseudomonas aeruginosa*, and *Acinetobacter baumannii* [18,25]. Notably, *blaNDM-1* often coexists with the *mcr-5.1* gene, which grants resistance to colistin, further complicating therapeutic choices [18,25].

### 3.2. 3rd Generation Cephalosporin Resistance Related Genes

*ESBLs* are enzymes that are responsible for the resistance of certain bacteria against a wide range of beta-lactam antibiotics, such as penicillins and cephalosporins. *ESBL*-producing bacteria, particularly *Escherichia coli* and *Klebsiella pneumoniae*, are frequently detected in hospital wastewater and are recognized as a major cause of healthcare-associated infections [20]. The release of *ESBL* genes into wastewater can raise resistance levels within community settings, underscoring the importance of wastewater as an AMR hotspot. The notable presence of *ESBL* genes in hospital wastewater acts as an early warning sign for potential outbreaks, given that the resistant strains can easily propagate through environmental and water systems [19].

Among the genes associated with extended-spectrum *beta-lactamase* resistance, several have been identified. The most common are *bla_CTX-M*, *bla_SHV*, *bla_TEM* and *bla_OXA-ESBLs* [27,28]. The pooled prevalence of ESBL Enterobacteriaceae in wastewater was found to be 24.81% (95% CI, 19.28–30.77) in a metanalysis that included 57 observational studies, with *bla*CTX-M genes being the most prevalent (66.56%). The results of this metanalysis also showed that hospital wastewater may be a significant repository of *ESBL* genes [29]. Furthermore, genes *blaNdm-1* and *blaCtx-m-15* have been linked to mobile genetic elements (MGEs), accelerating their horizontal transfer between environmental and clinical bacterial populations [18].

### 3.3. Colistin Resistance Related Genes

The identification of the *mcr* gene, conferring resistance to colistin, has generated significant concern in the medical field. Resistance to colistin is mediated primarily via the modification of the target site where colistin can make electrostatic interactions with Lipid A of lipopolysaccharides of the outer membrane [30]. Colistin, often considered as the last line of defense against multidrug-resistant Gram-negative bacterial pathogens, is unfortunately losing its effectiveness due to the emergence of *mcr* genes, particularly *mcr-1* and its variants, which have also been detected in hospital wastewater [31]. The *mcr-1* gene was described for the first time in China and subsequently various *mcr-1* allelic variants and *mcr*-type genes were identified. *Mcr-1* is the most prevalent with the *mcr-1.1* allelic variant being the most distributed worldwide [32]. *E. coli* is the main bacterial species carrying the *mcr-1* gene, a phenomenon that can be attributed to the ubiquity of this species, the presence of *mcr-1* in conjugative plasmids, and the extensive use of colistin in livestock [32]. However, *mcr* genes have also been found in *Klebsiella pneumoniae* and other *Enterobacteriaceae*, *Salmonella species*, *Aeromonadaceae*, and *Pseudomonas species* [3,17,33]. The ability of *mcr* genes to transfer between bacterial species aggravates the complexity of managing infections. In hospital wastewater, these genes are often associated with the selective pressure from antibiotic use, highlighting the critical need for strict monitoring and implementation of control practices [34]. Of particular concern is the presence of *mcr-5.1* in wastewater, signaling the potential for multidrug-resistant and pan-drug-resistant (PDR) bacteria in hospitals, posing a severe risk to public health [23].

### 3.4. Vancomycin Resistance and Methicillin Resistance Related Genes

*MRSA* is the cause of both community-onset and hospital-associated infections and has been linked to outpatient visits, long-term care facilities, antibiotic exposure, chronic illness, intravenous drug use, and close contact with individuals carrying these risk factors [35]. The primary determinant of methicillin resistance in *S. aureus* is the *mecA* gene, which encodes an altered penicillin-binding protein with a low affinity for β-lactam antibiotics (i.e., methicillin, oxacillin). *MecA* (and the newer variant *mecC*) is carried on a large mobile genetic element called *SCCmec* (Staphylococcal Cassette Chromosome mec), allowing horizontal transfer of methicillin resistance between staphylococcal species [36]. In a study conducted in both Greece and Romania, the analysis of staphylococcal genes in *MRSA* strains within the Greek patients revealed the presence of three key genes: *mecA*, *FemB*, and *SCCmec* elements [37].The *mecA* gene, the hallmark determinant of MRSA, has been detected in both hospital effluents and municipal wastewater, demonstrating the release of resistant pathogens into the environment [38].

Vancomycin resistance, primarily due to the *van* gene, presents significant challenges, especially in managing infections caused by *Staphylococcus aureus*. More than 100,000 deaths are caused by *MRSA* worldwide and at least 5400 deaths by *VRE* in the United States annually [39]. The presence of *VRE* in hospital wastewater represents a serious public health threat, as these organisms can rapidly spread through environmental pathways [3]. Research indicates that *VRE* can survive in water for extended periods, enabling its transmission to other pathogens and AMR spread [12]. Vancomycin resistance genes, notably *vanA*, have been identified in *Enterococcaceae*, *Staphylococcaceae*, underscoring the need for environmental monitoring to track resistance and guide infection control practices [15,19].

### 3.5. Sulfonamide Resistance Genes

The extensive use of sulfonamides, which act by inhibiting bacterial folic acid synthesis, has driven selective pressure, leading to the spread of sulfonamide resistance genes, including *sul1*, *sul2*, and *sul3*, in healthcare environments [15,17]. These genes, commonly detected in hospital wastewater, can enable isolates to bypass the drug’s action, leading to bacterial survival despite sulfonamide exposure. The persistence of *sul* genes in hospital wastewater points to a substantial burden of sulfonamide resistance, frequently associated with high antibiotic usage in clinical environments [40,41].

### 3.6. Tetracycline Resistance Genes

Resistance to tetracyclines, mediated by genes such as *tetA*, *tetB*, and most commonly *tetM*, is also widely detected in hospital wastewater [17]. Tetracyclines disrupt bacterial protein synthesis, yet the presence of *tet* genes in wastewater suggests selective pressure from the overuse of these antibiotics, raising concerns regarding their potential horizontal transfer to other clinically related pathogens [42,43].

## 4. Methods of ARGs Identification in Hospital Wastewater

Rapid detection of AMR traits is crucial to achieve guided treatment and limit the spread of resistant microorganisms. Various technical methods have been developed to identify resistance genes (ARGs), utilizing both culture-based and culture-free approaches, mainly molecular techniques and sequencing technologies [9].

### 4.1. Culture-Based Approaches

Culture-based techniques have been crucial in revealing AMR within hospital wastewater. These methods involve isolating and cultivating microorganisms from wastewater samples to isolate resistant strains and detect their resistance mechanisms. Standard isolation procedures include the use of selective media and enrichment cultures, with the last ones being used to enhance the population density of a particular group of microorganisms within the total microbial population of a sample [44]. Minimum inhibitory concentration (MIC) defines in vitro levels of susceptibility of specific strains and helps determine microbial isolate susceptibility to drugs, offering essential information on AMR [45]. Although accurate, these methods lack detection of non-culturable bacteria, which account for about 97% of the environmental microbiome. These techniques have been crucial in identifying pathogens like *Klebsiella pneumoniae*, *Escherichia coli*, and *Enterococcus* spp., which often have high levels of resistance [46]. Despite their value, culture-based methods are narrowed by their limitations, such as the potential to overlook non-culturable or fastidious organisms. Additionally, the presence of resistant bacteria in complex microbial communities can make isolation more puzzling [47], and thus these methods are gradually replaced by molecular techniques.

### 4.2. Polymerase Chain Reaction (PCR)-Based Methods

To address the limitations of culture-based methods for detecting antimicrobial resistant pathogens and phenotypes, molecular techniques like PCR and microarrays have been developed to augment AMR detection in microbes [46]. PCR is a technique that amplifies specific DNA sequences through repeated cycles of heating and cooling, leading to the creation of millions of copies of a targeted DNA fragment. Its advantages include speed, sensitivity, and specificity for detecting known mutations or pathogens, but it also has limitations as it only analyzes a small number of target sequences per reaction [48]. Its simplicity and reliability make it popular in diagnostics and genetic testing. PCR remains the key method for detecting antimicrobial resistance genes (ARGs), as it obtains both high sensitivity and specificity. Real-time PCR is currently one of the most powerful methods for DNA amplification. Various PCR techniques, including conventional PCR, qPCR, and multiplex PCR, are used to isolate known ARGs in clinical samples. For instance, multiplex PCR enables simultaneous detection of multiple ARGs, which is particularly useful in multi-drug resistance cases. Additionally, qPCR can provide semi-quantitative data by evaluating bacterial load and gene expression levels, offering insights into gene responses after their exposure to antibiotics [14,34,42]. For example, specific genes like *blaCTX-M*, *blaKPC*, *qnrB*, and *mcr-1* can be easily detected with the use of qPCR [49]. Although qPCR has many advantages by being a rapid, cost-effective, and sensitive method for detecting and quantifying ARGs, its application relies heavily on known target sequences that contain conserved primer sites [42]. Reverse-transcription PCR (RT-PCR), which targets RNA instead of DNA, can also be utilized to examine gene expression. Moreover, droplet digital PCR (dd-PCR) enables precise absolute quantification and demonstrates strong resistance to inhibitors, making it a preferred alternative to address the limitations of qPCR [50]. While PCR is effective for detecting known ARGs, it has limitations when it comes to identifying novel resistance mechanisms [51,52]. Last, although PCR-based methods are highly sensitive in detecting low-abundance genes, they are vulnerable to mutations in the primer target regions, which can lead to false-negative results [42].

### 4.3. Microarray Technology

Microarrays are essential tools for detecting multiple antimicrobial resistance genes (ARGs) in a single test. This technology quantifies and identifies AMR genes and their expression patterns by hybridizing probe DNA/RNA sequences to complementary target sequences on a microarray plate [46]. It is particularly effective for the simultaneous multiplex detection of various ARGs [53], while its rapid performance and suitability for epidemiological or infection control studies that require the characterization of large sets of isolates makes it extremely useful [54]. By analyzing the patterns of hybridization, microarrays can identify the existence of several resistance genes, especially those that are associated with *β-lactamases*, aminoglycosides, and tetracyclines. These systems are promising when it comes to genotyping, boasting high multiplexing capabilities that allow the detection of multiple *β-lactamase* genes within a single strain. Microarray systems like the Affymetrix GeneChip^®^ and Check-Points^®^ microarray are commonly employed for AMR surveillance in both environmental and clinical settings [34,55].

On the other hand, microarrays entail limitations that make them less than ideal for broad-scale wastewater surveillance. A primary challenge is their reliance on known genetic sequences for effective detection, meaning they can miss pathogens or ARGs that lack comprehensive genomic information. Additionally, wastewater samples contain various inhibitors and complex particulates, which can interfere with the accuracy of microarray detection. Sample processing and nucleic acid extraction steps can lead to data inconsistency, impacting the sensitivity and specificity needed for reliable results [14,56]. These systems are highly efficient and capable of processing a large number of samples, but their main limitation is that, while they excel in throughput, they are not as effective in discovering previously unrecognized resistance genes [57].

### 4.4. Next-Generation Sequencing Techniques

NGS technologies, and particularly whole-genome sequencing (WGS), offer extensive insights into AMR by enabling the detection of both established and novel ARGs. NGS is a high-throughput technology that sequences millions to billions of DNA fragments simultaneously and can examine entire bacterial genomes, allowing for the identification of mutations linked to resistance mechanisms [23,58]. It provides comprehensive genomic, transcriptomic, and epigenetic data, allowing for broad discovery and detailed analysis of complex genetic variations, unknown mutations, and large-scale genomic patterns [59]. Another approach, targeted sequencing, focuses on specific genome regions likely to contain ARGs. Sequencing technologies such as Illumina^®^ and Ion Torrent^®^ allow for the rapid sequencing of pathogens directly from clinical samples, eliminating the need for culturing and thus accelerating diagnosis [3,25]. Furthermore, Nanopore^®^ sequencing (e.g., the MinION platform) provides real-time sequencing capabilities, with tailored workflows and improving accuracy [21]. NGS is scalable, producing vast amounts of data for whole-genome, exome, or targeted segment sequencing, but generally takes longer and is more resource-intensive than PCR. In a study comparing high-throughput quantitative PCR (HT qPCR) and metagenomic sequencing for ARG screening in hospital wastewater, metagenomics offered a broad view of the resistome and was instrumental in evaluating ARG risk. Conversely, HT qPCR showed higher sensitivity in quantifying all targeted and clinically relevant ARGs [21]. Currently, the use of NGS is primarily for microbial strain typing for epidemiological purposes; however, its potential to provide clinically valuable insights through microbial community analysis is increasingly recognized [60]. The advantages and disadvantages of all the above techniques described are outlined in Table 2.

Hospital drains and water interfaces are implicated in the nosocomial transmission of pathogens [55]. The detection of antimicrobial resistance genes relies on a diverse array of technologies. Culture-based approaches are a reliable method for providing proof of viable bacteria of concern, but they have several limitations [9]. PCR-based methods remain foundational for identifying known ARGs, while next-generation sequencing can provide more large-scale, in-depth genomic exploration. By integrating these methods into routine diagnostic workflows and environmental surveillance systems, it could facilitate a more comprehensive management of AMR globally.

## 5. The Role of Hospital Wastewater Surveillance in Antimicrobial Resistance

Wastewater-based surveillance (WWS) has emerged as a transformative approach in public health epidemiology and can also be applied in monitoring AMR in healthcare settings. WWS captures resistome signals from the combined output of hospital activities and offers population-scale insights into AMR burden. Wastewater is believed to play a critical role in the dissemination of AMR due to the abundance of bacteria, including resistant ones, the potential for exchange of genetic elements between them, the dynamic confluence of water from different sources, and the AMR dynamics of communities, healthcare facilities, and industry [3]. Therefore, WWS programs are increasingly recognized as significant tools in monitoring AMR, helping mitigate outbreaks, raising awareness, and providing valuable data for further research. These programs provide a population-level snapshot of antimicrobial use and resistance patterns, complementing traditional hospital- and clinic-based surveillance systems.

In healthcare settings, home to infections by resistant pathogens and high-level antimicrobial consumption, specific environmental conditions can support the persistence and spread of resistant bacterial strains (Figure 1). Several studies have highlighted the benefits of HWWS for AMR monitoring and control. HWWS can serve as an indicator of local AMR burden, reflecting the resistance profiles of hospital-acquired infections [19]. In a study at Western General Hospital in Edinburgh, metagenomic analysis and multiple-site sampling were used to investigate the relationship between antimicrobial resistance gene prevalence and clinical activities, such as antimicrobial usage and hospitalization duration [19]. Samples were collected over 24 h from various wastewater collection points within a tertiary hospital and compared with community sewage. Results indicated a higher abundance of ARGs in hospital wastewater compared to community influents. Although total antimicrobial usage did not show a direct correlation with elevated ARG levels, a modest positive association was observed with certain ARG phenotypes. Furthermore, increased use of carbapenem and vancomycin was positively linked to the presence of ARGs associated with these specific antibiotics. Notably, *Enterococcaceae* and *Staphylococcaceae* isolates were the most prevalent [19]. Overall, analyzing sewage samples offers a cost-effective method for monitoring antibiotic-resistant pathogens at a population level. A study conducted in Gothenburg, Sweden, revealed that *E. coli* resistance rates derived from hospital sewage and hospital patients strongly correlated, as did resistance rates in *E. coli* from municipal sewage and primary care urine samples [54]. This approach could complement existing monitoring systems by addressing challenges related to limited sampling in clinical practice.

Moreover, wastewater surveillance can capture both pathogenic and non-pathogenic resistant bacteria, offering a comprehensive assessment of resistance gene reservoirs, including ARGs that may otherwise go undetected by following clinical surveillance alone [9,61]. Hospital wastewater surveillance can also help monitor clinically significant pathogens and thus contribute to the early identification of outbreaks. The approach was successfully employed in large-scale wastewater-based surveillance during the COVID-19 pandemic, where it proved valuable in estimating infection prevalence and predicting outbreaks based on genetic material detected in wastewater [62,63]. To address limitations impacting signal accuracy, such as wastewater composition variability and dilution from other effluents, near-source sampling (NSS) was employed. A study at the University of Arizona demonstrated that, when NSS was correlated with clinical samples, it provided a sensitive approach to identify acute cases within a defined population, such as hospital patients. This method could thus be valuable for monitoring shifts in disease dynamics and providing early warning of potential site-specific outbreaks [62]. Similar approaches for AMR monitoring could enhance early warning capabilities for AMR in healthcare facilities, thereby improving public health responses to potential future outbreaks [3].

Table 3 summarizes some recently published studies of antimicrobial resistance gene distributions in hospital wastewater in different hospital types and geographic regions, with sampling covering both raw and treated hospital wastewater. The key antimicrobial resistance genes identified across these studies include *carbapenemase* genes, colistin resistance determinants, and extended-spectrum *beta-lactamase* genes [3,6,14,15,16,17,18,19,20].

Despite its potential, HWWS faces limitations, including the lack of standardized guidelines and protocols for sample collection and analysis. Variations in sampling and analysis methods, timescales, and geographic locations can yield inconsistent results that are difficult to compare [9]. Notably, when comparing antimicrobial resistance data from hospital wastewater treatment plants across different countries, several factors should be taken into consideration, like differences in environmental regulations, variable treatment technologies, and allocated financial resources that can have an impact on the efficiency of ARG removal and AMR containment. Moreover, HWWS alone cannot differentiate the exact sources of contamination within the healthcare facility. In addition, ARGs in wastewater could stem from non-human sources, such as veterinary settings or environmental reservoirs [17]. Developing standardized protocols and harmonized procedures is therefore essential to enable comparable findings and to strengthen the role of WWS in managing the persistent threat of resistant pathogens [14].

## 6. Role of Hospital Wastewater Treatment Plants

Wastewater treatment plants (WWTPs) are designed to sequentially remove contaminants, including pathogenic microorganisms and organic matter, from wastewater before environmental discharge [66]. Hospital WWT systems are constituted by several stages designed to minimize organic matter, microorganisms and chemical contaminants. Initial stages remove solids through screening and sedimentation, followed by biological treatment processes, such as biofilm reactors, that degrade organic pollutants via microbial metabolism. Finally, disinfection steps, commonly including chlorine, ultraviolet radiation, or advanced oxidation, are used for inactivation of the pathogenic organisms and reduction in microbial loads [15,17].

Despite being critical control points aimed at reducing bacterial load, WWTPs can also act as reservoirs and even amplifiers of antimicrobial resistance. Despite the above treatment steps, AMR can persist in treated effluents for many reasons. First, several resistant bacteria have the ability to adapt to treatment stressors, or they survive partial inactivation. Secondly, extracellular DNA, which contains ARGs may not be effectively removed, and therefore horizontal gene transfer to other bacteria in the environment can be facilitated. Furthermore, residual antimicrobials present in hospital wastewater exert selective pressure, resulting in the maintenance and spread of resistance determinants [3,19]. Additionally, variation in treatment plant designs and operational parameters influences ARG removal efficiency. Conventional chlorination, one of the most common disinfection methods, is optimized to inactivate a broad spectrum of pathogens but does not specifically target or fully eradicate ARBs and extracellular ARGs. In certain contexts, chlorination may paradoxically facilitate persistence or even the enrichment of resistant bacteria and genes within treatment systems and downstream effluents. This can occur because chlorine disinfection is less effective against bacteria protected within biofilms or sludge aggregates and fails to degrade extracellular DNA harboring ARGs, thereby maintaining the environmental reservoir for horizontal gene transfer [26,67]. Alternative or complementary treatments—such as ultraviolet disinfection, ozonation, membrane filtration, and advanced oxidation processes—offer varying degrees of ARG removal, yet their adoption depends on resource availability and regulatory frameworks, which differ globally.

Hence, hospital wastewater treatment plants (HWWTPs) have been identified as an emerging source of antimicrobial-resistant bacteria (ARB) that might cause horizontal gene transfer among microbial communities in water systems [3]. Numerous studies have identified ARGs in various water sources originating from WWTPs. Indeed, a study in Mexico found carbapenemase-producing Klebsiella species surviving WWTP processes, highlighting the role these plants may inadvertently play in AMR dissemination [20]. Meng Y. et al. describe in their article how they employed metagenomic sequencing of both influent and effluent hospital wastewater to assess bacterial composition and resistome. The results revealed that most ARGs decreased after treatment, but a subset persisted or increased in effluent [64]. Another study, which was conducted in India, involved 24 hospital wastewater samples that were collected over a period of 6 months. In these samples, Gram-negative bacteria were cultured and identified, and antimicrobial resistance was tested, revealing high rates of carbapenem and cephalosporin resistance in the final treated effluent [65]

Consequently, comprehensive monitoring of hospital WWTPs is essential to understand transmission dynamics of ARGs and to inform the design of enhanced treatment strategies. There is an urgent need for innovation towards the optimization of WWTP designs specifically targeting the effective mitigation of antimicrobial resistance risks associated with hospital wastewater discharges.

## 7. Future Perspectives

AMR remains a mounting global threat, necessitating coordinated, urgent action [66]. Wastewater has been shown to play an integral role in AMR development, as it serves as a substantial reservoir for ARBs and ARGs [9], with a significant correlation with human AMR prevalence, as shown in various studies, regardless of their heterogenous approaches [12]. Prioritizing the development of standardized methodologies, alongside regulatory frameworks and interdisciplinary collaboration on wastewater AMR surveillance, will enable comparability, data sharing, and actionable reporting across regions [9]. Parameters that should be included in these protocols are specifications regarding the time/location-matched sampling of wastewater and human populations, compound influent and effluent sampling, while sampling collection for >12months could provide data for longitudinal studies [12]. Future research should focus on the validation and harmonization of sampling, data analysis, and reporting protocols; the integration of clinical, epidemiological, and environmental datasets; and the advancement of rapid, cost-effective surveillance platforms. The advancement of wastewater surveillance could support timely, evidence-based decision-making, enhance our ability to control AMR dissemination, enable the development of better treatment methods, and contribute to the preservation of antimicrobial efficacy for patients and communities worldwide.

## 8. Conclusions

Wastewater-based surveillance is now recognized as an important public health tool. Hospital wastewater-based surveillance has the potential to serve as a critical tool for understanding, tracking, and potentially mitigating the spread of AMR in highly endemic settings. By enabling comprehensive, real-time monitoring of ARGs and resistant bacteria from concentrated clinical sources, hospital wastewater surveillance provides a platform that closely reflects local prescribing practices and clinical AMR patterns. Unlike traditional surveillance methods limited to symptomatic patients, this approach captures data on asymptomatic carriage, transmission dynamics, and emerging resistance mechanisms within institutions, complementing clinical datasets and reducing reporting bias. Evidence so far has demonstrated significant correlations between ARG profiles in hospital influent and regional clinical data, supporting the use of wastewater analysis for targeted interventions, outbreak detection, and evaluation of stewardship policies.

## Figures and Tables

**Figure 1 microorganisms-13-02739-f001:**
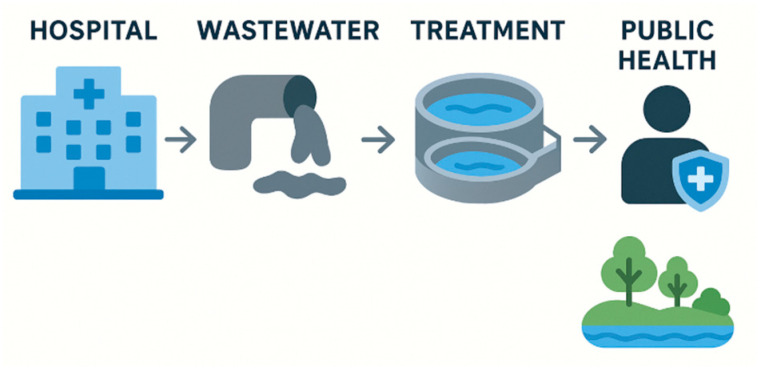
The spread of antimicrobial resistance genes from Hospital Wastewater.

**Table 1 microorganisms-13-02739-t001:** The most common ARGs detected in hospital wastewater.

Antibiotic Class	Key Resistance Genes	Mechanism
Carbapenems	*bla_KPC*, *bla_NDM*, *bla_VIM*, *bla_OXA-48*, *bla_IMP*	*Carbapenemase* production leading to β-lactam degradation
B-lactams	*bla_CTX-M*, *bla_SHV*, *bla_TEM*, *bla_OXA-ESBLs*	Hydrolyze third-generation cephalosporins
Colistin	*mcr-1 to mcr-10*, *pmrA*, *pmrB*	Modification of lipid A reducing colistin binding
Tetracyclines	*tetA*, *tetB*, *tetM*, *tetO*, *tetX*, *tetC*, *tetD*, *tetG*	Efflux pumps and ribosomal protection proteins
Sulfonamides	*sul1*, *sul2*, *sul3*	Altered dihydropteroate synthase
Vancomycin	*vanA*, *vanB*, *vanC*, *vanD*, *vanE*, *vanG*, *vanL*, *vanM*, *vanN*	Modified peptidoglycan precursors reducing binding

ARG = antimicrobial resistance genes; ESBL = extended-spectrum *β-lactamase*.

**Table 2 microorganisms-13-02739-t002:** Comparison between the techniques used to identify ARGs and monitor hospital wastewater surveillance.

Technique	Advantages	Disadvantages	Key Applications in Hospital Wastewater Surveillance
**Culture-Based Approach**	Reliable for identifying viable bacteria which have clinical significance (proof of existence)	Unable to detect non-culturable bacteria (only ~1–3% of environmental bacteria can be cultured)	Useful for detailed susceptibility testing and resistance pattern analysis
	Enables susceptibility testing to determine resistance patterns	Bacteria often die quickly when exposed to non-native environmental conditions	May miss non-culturable or difficult-to-culture organisms
**qPCR (Quantitative PCR)**	High sensitivity and specificity, detects hard-to-culture bacteria	Requires prior information on target sequence for assay design	Effective for rapid detection of target ARGs, even at low concentrations
	Can detect low-abundance ARGs, allowing early outbreak detection	Limited to preselected ARGs, unable to detect non-targeted or novel ARGs	Suitable for focused surveillance of known resistance genes in wastewater
**Metagenomics**	Detects a wide range of ARGs, including novel and divergent genes	More expensive per sample, especially if applied in small-scale applications	Comprehensive analysis for total ARG burden, capturing more gene diversity
	More cost-effective for large sample monitoring	May be less sensitive for low-abundance ARGs compared to targeted methods	Ideal for broader surveillance across multiple resistance genes and bacteria

**Table 3 microorganisms-13-02739-t003:** Examples of recently published studies of antimicrobial resistance gene detection in hospital wastewater.

Study	Country (City)	Hospital Type	Number of Beds (If Reported)	Sampling Site	Key ARGs Detected
Meng et al., 2025 [64]	China	Tertiary university hospital (Sir Run Run Shaw Hospital, Hangzhou, Eastern China)	Not reported (but HWTS processes ~10,000 tons/week, so likely large)	Influent and effluent of hospital wastewater treatment system (HWTS)	Beta-lactam (e.g., *blaCTX-M*, *blaPER*, *cfxA3*, *blaKPC*, *blaVIM*, *blaNDM*, *blaIMP*), aminoglycoside, macrolide, phenicol, tetracycline, vancomycin (*vanH-M*, *vanX-M*, *vanM)*, linezolid *(cfr(C)*, *optrA*), colistin (*mcr-4.3*, *mcr-5.1*), tigecycline (*toprJ*, *toprJ1*, *tetX2*), QAC resistance genes *(qacF*, *qacG2*)
Shetty et al., 2025 [65]	India	Not specifically reported	Not reported	Hospital wastewater samples, including final treated effluent	*tetD*, *tetE*, *tetG* (tetracycline resistance), *catA1*, *catA2* (chloramphenicol resistance), *blaNDM-1* (carbapenem resistance), *qnrA*, *qnrB*, *qnrS* (quinolone resistance), *qepa*
Galarde-López et al., 2024 [14]	Mexico (Mexico City Metropolitan Zone)	Two Tertiary hospitals	Not reported	Hospital wastewater effluent	*Bla-KPC*, *bla-NDM*, *mcr-1*, *bla-Oxa48-like*
Stoesser et al., 2024 [16]	United Kingdom (Edinburgh)	Tertiary Teaching Hospital	Not reported	Multiple wastewater collection points within hospital wards	*bla_KPC*, *bla_NDM*, *blaIMP*, *blaVIM*, *blaOXA-23-like*, *blaOXA-48-like*, *blaOXA-58-like*
Galarde-López et al., 2024 [20]	Mexico	Two hospital wastewater treatment plants serving general hospitals	Not reported	Hospital wastewater treatment plant effluents	Carbapenem resistance genes (e.g., *bla_KPC*, *bla_NDM*) Extended-spectrum *β-lactamase* (*ESBL*) genes Colistin resistance genes (*mcr-1* and variants) Other multidrug resistance genes found in clinical isolates of *E. coli*, *Enterobacter* spp., and *Acinetobacter* spp.
Delgado-Blas et al., 2022 [18]	Ghana	General Hospital (urban hospital wastewater canalizations)	400 (approximately)	Hospital and urban wastewater canalizations	*bla_KPC*, *bla_NDM*, *bla_VIM*, mcr variants, including mcr-1 found coexisting with *bla_NDM-1*, *ESBL* genes
Siri et al., 2021 [3]	Thailand	General hospital	Not reported	Hospital wastewater and related wastewater treatment plant effluents	Carbapenem resistance genes (e.g., *bla_KPC*, *bla_NDM*), Extended-spectrum *β-lactamase* genes (*bla_CTX-M*, *bla_SHV*, *bla_TEM*), Colistin resistance genes (*mcr* genes, primarily *mcr-1*)
Perry et al., 2021 [19]	Scotland (Edinburgh)	Tertiary Teaching Hospital	Not reported	Hospital drains and wards wastewater collection points	Carbapenem resistance genes: *bla_KPC*, *bla_NDM*, *bla_IMP*, *bla_VIM*, *bla_OXA-23*, *bla_OXA-48*, *bla_OXA-58*, *ESBL genes: bla_CTX-M*, *bla_SHV*, *bla_TEM*, Colistin resistance genes: *mcr-1 to mcr-10* Vancomycin resistance genes: *vanA*, *vanB*, *vanC*, etc., Tetracycline resistance genes: *tetA*, *tetB*, *tetM*, etc., Sulfonamide resistance genes: *sul1*, *sul2*, *sul3*, Methicillin resistance gene: *mecA*

Many of the included studies did not specify the exact number of hospital beds; where specific data are lacking, this is indicated. Sampling sites included raw effluent, treated wastewater, treatment plants, hospital drains, and specific wards.

## Data Availability

No new data were created or analyzed in this study. Data sharing is not applicable to this article.

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
