# Peer review of "Hospital Wastewater Surveillance and Antimicrobial Resistance: A Narrative Review"

_microorganisms, 2025, doi:10.3390/microorganisms13122739_

Round 1

Reviewer 1 Report

Comments and Suggestions for Authors

Paper is a well-written and informative overview of the importance of wastewater surveillance in addressing antimicrobial resistance (AMR). It effectively explains the role of advanced detection methods such as PCR, NGS, and microarray technologies. The transition between community and hospital settings is logical and enhances the clarity of the discussion. The connection between surveillance findings, infection control, and antibiotic stewardship is particularly strong and relevantly emphasized. To further strengthen the work, adding specific and up-to-date references (e.g., WHO, CDC, or key studies from 2023–2025) would improve its academic rigor.

Including citations for examples of key ARGs—such as ESBLs, carbapenemases, colistin, and vancomycin resistance—would also increase credibility.

Minor changes. Paper effectively communicates the global significance of AMR monitoring.

Reviewer 2 Report

Comments and Suggestions for Authors

The manuscript presents the management of wastewater in the hospital and how it relates to the antimicrobial resistance.  The manuscript contains very general information and does not have much significance nor novelty. The authors need to elevate the quality of data or findings presented in the manuscript. Here are several comments for the manuscript:

  1. This manuscript is a review not article.
  2. There is no specific aims in the abstract nor mentioned in detail in introduction.
  3. Hospital wastewater term needs clarification and definition.
  4. The abbreviation of genus name can only be done after full mentioned of the name at the beginning. Look after this mistake the whole manuscript.
  5. Materials and method section is not relevant. There is no systematic and metadata analyzed in this study.
  6. PubMed database needs clarification, why it was chosen and when the access done.
  7. Table 1 is poorly presented, revise and add more information.
  8. What is the difference between PCR and next gen sequencing?
  9. Section 5 and 6 are poorly constructed, there is no specific explanation on how WWT works and why there is still chance of AMR happens. Why did the authors consider the chlorine treatments? or any other biological treatment to basically reduce the potential hazard.

The authors need to carefully address these comments and also it needs a lot of advancing in terms of discussion and depth of the data analysis. There is no figure or flow chart to support the explanation as well. 

Reviewer 3 Report

Comments and Suggestions for Authors

1. The abstract needs to be rewritten. Now the abstract in terms of content is an introduction or even a conclusion. The abstract should introduce us to what we will see in the article. 2. I don't have enough illustrations in the text. 3. It is necessary to indicate the hospitals for how many beds from which countries were included in the study. I think it can be shown in the form of a table or a map. 4. How correct is it to compare data on wastewater treatment plants from different countries at the same time? Is there a different regulatory framework everywhere? Again, if we take the EU and African countries, they have different financial capabilities. Very controversial. 5. I don't have enough specifics in the text, the authors often go into populism.

Round 2

Reviewer 2 Report

Comments and Suggestions for Authors

The manuscript has been revised and the comments have been replied accordingly. Nevertheless, the quality of the manuscript still does not meet the standard for acceptance. There is a lack of depth and specific discussion to address actual problem in WWT process. The novelty and comprehensive ideas from the authors are also missing. For this reason, I suggest rejection for the manuscript and will not accept any invitation for further corrections.

Author Response

Comment #1

The manuscript has been revised and the comments have been replied accordingly. Nevertheless, the quality of the manuscript still does not meet the standard for acceptance. There is a lack of depth and specific discussion to address actual problem in WWT process. The novelty and comprehensive ideas from the authors are also missing. For this reason, I suggest rejection for the manuscript and will not accept any invitation for further corrections.

Answer to Comment #1

We would like to express our sincere appreciation to the reviewer for his/her time and effort in evaluating our revised manuscript. We fully respect the reviewer’s opinion and acknowledge their careful consideration; however, we must respectfully disagree with the conclusion that the work lacks depth and sufficient discussion of wastewater treatment processes.

In our revision, we substantially expanded the discussion to address specific technical aspects of the wastewater treatment process and to integrate recent literature that contextualizes our findings within the broader framework of antimicrobial resistance dissemination. These additions were made precisely to enhance the manuscript’s depth and relevance.

Reviewer 3 Report

Comments and Suggestions for Authors

64-71 lines. I'm not insisting, but I think these lines are unnecessary.

line 124. There are strange links here and further along the text, all separately. Wouldn't it be better to combine them and do [3, 10, 18-24]? Similarly, down the text.

line122. The authors introduce the abbreviation ESBL. 128 line. The authors once again introduce the abbreviation ESBL. Section 3 is sloppily written.

134-135 «Kpc-producing bacteria represent a real challenge in clinical environments due to their ability to degrade approximately all beta-lactam antibiotics, including carbapenems, posing significant treatment limitations (25)» What does it mean that a bacteria …. ability to degrade, a a bacteria does not degrades anything. It is not scientifically written, it needs to be reformulated.

Table1, line 2. Extended-spectrum β-lactamases is an enzyme, not an antibiotic class, the antibiotic is β-lactam!!!

Table 2. Please add the cost of the procedure, the cost of the equipment required for the behavior of the technique, the duration of the analysis in time.

Right now, the Conclusions are very empty and insignificant, so I suggest going around the Conclusions section with the Future perspective section.

Author Response

We thank the reviewer for the meaningful and helpful comments and recommendations.

Comment #1

64-71 lines. I'm not insisting, but I think these lines are unnecessary.

Answer to Comment #1

After the reviewer’s comment we have decided to remove the above lines from our manuscript.

Comment #2

There are strange links here and further along the text, all separately. Wouldn't it be better to combine them and do [3, 10, 18-24]? Similarly, down the text.

Answer to Comment #2

Thank you very much from this comment. Due to a technical challenge with the citation manager platform Zotero, in order to avoid any citation errors, we have kept the references in the output form provided and are willing to further edit at a later stage of the submission, if our manuscript is accepted for publication.

Comment #3

line122. The authors introduce the abbreviation ESBL. 128 line. The authors once again introduce the abbreviation ESBL. Section 3 is sloppily written.

Answer to Comment #3

We would like to thank the reviewer for his/her comment. We have now corrected the abbreviation mistake in our manuscript.

Section 3 has now been revised in order to enhance clarity, content and form.

Comment #4

134-135 «Kpc-producing bacteria represent a real challenge in clinical environments due to their ability to degrade approximately all beta-lactam antibiotics, including carbapenems, posing significant treatment limitations (25)» What does it mean that a bacteria …. ability to degrade, a  bacteria does not degrades anything. It is not scientifically written, it needs to be reformulated.

Answer to Comment #4

We would like to thank the reviewer for his/ her comment. We have now reformulated this sentence so that it is scientifically accurate.

Comment #5

Table1, line 2. Extended-spectrum β-lactamases is an enzyme, not an antibiotic class, the antibiotic is β-lactam!!!

Answer to Comment #5

We would like to thank the reviewer for his/her constructive observation. We have now corrected this mistake in Table 1.

Comment #6

Table 2. Please add the cost of the procedure, the cost of the equipment required for the behavior of the technique, the duration of the analysis in time.

Answer to comment #6

We appreciate the reviewer’s suggestion to include cost and duration data. However, the primary aim of our article is to review and analyze antimicrobial resistance gene detection methodologies and AMR related implications in hospital wastewater. Detailed economic and operational cost analyses were beyond the scope of our work, as such parameters vary significantly between settings, platforms, countries and for some cases, operator technical experience, and require specialized study. We suggest that these important aspects could be explored in future dedicated investigations.

Comment #7

Right now, the Conclusions are very empty and insignificant, so I suggest going around the Conclusions section with the Future perspective section.

Answer to comment #7

We would like to thank the reviewer for his/her comment. We have revised the conclusion section and made it more informative, including significant points reviewed in the manuscript. We placed a special focus on future perspectives and we now believe that it has been ameliorated.
